# Role of microRNAs in Diagnosis, Prognosis and Management of Multiple Myeloma

**DOI:** 10.3390/ijms21207539

**Published:** 2020-10-13

**Authors:** Amro M. Soliman, Teoh Seong Lin, Pasuk Mahakkanukrauh, Srijit Das

**Affiliations:** 1Department of Biological Sciences—Physiology, Cell and Developmental Biology, University of Alberta, Edmonton, AB T6G 2R3, Canada; amsherif@ualberta.ca; 2Department of Anatomy, Faculty of Medicine, Universiti Kebangsaan Malaysia Medical Centre, Kuala Lumpur 56000, Malaysia; 3Department of Anatomy & Excellence in Osteology Research and Training Center (ORTC), Chiang Mai University, Chiang Mai 50200, Thailand; pmahanka@mail.med.cmu.ac.th

**Keywords:** multiple myeloma, cancer, miRNA, oncomiR, tumor suppressor, diagnostic marker

## Abstract

Multiple myeloma (MM) is a cancerous bone disease characterized by malignant transformation of plasma cells in the bone marrow. MM is considered to be the second most common blood malignancy, with 20,000 new cases reported every year in the USA. Extensive research is currently enduring to validate diagnostic and therapeutic means to manage MM. microRNAs (miRNAs) were shown to be dysregulated in MM cases and to have a potential role in either progression or suppression of MM. Therefore, researchers investigated miRNAs levels in MM plasma cells and created tools to test their impact on tumor growth. In the present review, we discuss the most recently discovered miRNAs and their regulation in MM. Furthermore, we emphasized utilizing miRNAs as potential targets in the diagnosis, prognosis and treatment of MM, which can be useful for future clinical management.

## 1. Introduction

Multiple myeloma (MM), also recognized as plasma cell myeloma, is a cancer of the plasma cells. MM develops in the bone marrow and originates from the long-lived plasma cells following their maturation in lymph nodes and migration to the bone marrow [1,2,3]. In most MM cases, the disease starts with pre-malignant asymptomatic stages, which eventually evolve to symptomatic intramedullary and extramedullary MM. A section of MM patients is pre-diagnosed with monoclonal gammopathy of undetermined significance (MGUS), which is found in around 3% of individuals over 50 years [4,5]. Nearly 1% of patients diagnosed with MGUS develop MM or other hematological malignancies [6,7,8]. Meanwhile, other patients are diagnosed with smoldering MM (SMM), an asymptomatic and more progressive pre-malignant stage, which could be detected through clinical examination and laboratory investigations [9]. Almost 10% of SMM cases develop into MM during the first five years subsequent to the diagnosis [10].

MM is the second most common blood cancer, where it represents 1% of all cancers and about 13% of all hematologic malignancies globally [11,12,13]. The incidence rate of MM in Europe is 5 per 100,000 [14,15]. In the USA, almost 230,000 cases of MM at different stages were reported from 2011 to 2016, with around 20,000 new cases registered yearly [13,16]. Approximately 28.6% of MM cases are diagnosed at the age of 65–74 years, and about 3.5% under the age of 44 years [17]. The incidence of MM in the black race is higher than the white race and is more common in males than females [18]. The median survival following standardized therapy is 3 to 4 years. However, bone marrow transplantation could expand survival to 7 years [19].

## 2. Pathogenesis of MM

The exact cause of MM remains unknown. However, a number of risk factors have been reported, including radiation, family history and obesity [20]. Furthermore, numerous genetic mutations are associated with MM development [21,22]. Approximately 90% of MM patients possess genetic defects in their plasma cells, which varies with the advancement of the disease. For instance, certain cytogenetic variations result in the evolvement of MGUS or SMM to MM [23]. These cytogenetic alternations include del(17p), t(11;14) (q13;q32), del(13), t(4;14) (p16;q32), t(14;20) (q32;q11), gain(1q), isolated monosomy 13, monosomy 14 and del(1p) mutations [24]. The cytogenetic alternations have a great impact on the prognosis, drug resistance and median survival of MM [24]. The molecular categorization of MM is carried out based on the cytogenetic mutations observed in plasma cells through fluorescence in situ hybridization studies. Most MM patients were associated with trisomies of one or more odd-numbered chromosomes detected in malignant plasma cells. Meanwhile, approximately 30% of MM cases were characterized by the translocation of the immunoglobulin heavy chain (IgH) locus on chromosome 14 [22,25,26]. Both trisomies and IgH translocations were reported in about 15% of MM cases [26].

## 3. Epigenetics of MM

The expanding literature is supporting the role of epigenetic aberrations in MM initiation and progression. These aberrations are detected at the onset or during the evolvement of the disease [22,27,28] and likely attributed to the significant variations in response to therapy and survival rates [29,30,31]. In addition to genetic alteration, the enduring research data indicate a potential role of epigenetic changes such as histone modification [22,32,33] and DNA methylation [34,35,36,37] in MM development. Mutations of specific epigenetic modifiers, including histone acetylation and histone methylation regulators, were reported in MM cases [38,39]. Intriguingly, epigenetic aberrations were found to regulate the expression of essential tumor suppressor miRNAs through hypermethylation [40,41,42]. Further, they seem to be the most critical underlying mechanism behind the dysregulation of miRNAs expression in MM [43,44,45,46]. On the other hand, miRNAs themselves can influence the epigenetic machinery by targeting their enzymatic modifiers and mediators [47]. For instance, miR-29b targets de novo methyltransferases mRNAs; thus, inhibiting DNA methylation in MM cells [48]. The epigenetic alterations further contributed to MM plasticity by enhancing the phenotypic diversity of MM cells and drug resistance [49,50].

DNA methylation is accomplished by adding a methyl group to the cytosine base, which can change the activity of DNA without changing its sequence. For instance, when methylation is present in a promoter region, it represses the expression of the gene. The methylation process is controlled by various enzymes such as DNA methyltransferases 1, 3A and 3B (DNMT1, DNMT3A and DNMT3B) in addition to the ten-eleven translocation (TET) protein family [51]. In MM, malignant cells are generally associated with DNA hypermethylation of tumor suppressor genes [51,52,53,54,55,56,57] and overall hypomethylation resulting in genomic instability [58,59], disease advancement [60], poor prognosis [34] and drug resistance [61]. Methylation of several tumor suppressor miRNAs such as miR-34 family, miR-194, miR-192 and miR-215 were identified in MM cases, which in turn resulted in their silencing [62,63,64,65,66].

Being an essential component of DNA chromatin, histone links the nucleosomes together and promotes the development of high ordered chromatin structures. The posttranslational modifications of the N-terminal tail of the histone protein affect gene transcription and DNA repair [67,68]. Therefore, aberrant posttranslational modifications of histones result in tumorigenesis [69]. Furthermore, the N-terminal tails are subject to methylation, acetylation and phosphorylation, impacting the gene transcription [67,69,70]. The emerging data are suggesting a critical role of histone posttranslational modifications in MM pathogenesis [71]. Altered posttranslational modifications were found to deregulate miRNAs in MM cells. For instance, miR-26 was down-regulated due to heterochromatin modification in t(4; 14) myeloma, which in turn led to enhanced MM cell proliferation [72].

## 4. Diagnosis and Management of MM

The diagnosis of MM depends mainly on both radiological and broad laboratory examinations. Symptoms revealed by MM patients are usually non-specific and of reduced value to confirm the diagnosis. Most of the MM cases presented with a history of anemia of unknown origin for an extended period. Additionally, they suffered from nausea, vomiting, generalized weakness, fatigue and weight loss [17]. Therefore, clinical examinations, in addition to extensive radiological and laboratory investigations, including complete blood count and cytogenic analysis, are crucial for an accurate diagnosis [17,73]. MM is commonly associated with monoclonal protein, abnormal immunoglobulin (Ig), production, including IgG and IgM [22]. Monoclonal protein can only be detected in around 82% of patients through serum protein electrophoresis [17]. However, performing the serum-free light chain assay or 24-h urinary protein electrophoresis with immunofixation increased the capacity of monoclonal protein detection [74]. It is worth mentioning that approximately 3% of MM patients have no reportable trace of monoclonal protein [75]. Abnormal monoclonal protein leads to blood hyperviscosity, end-organ failure, immunodeficiency in addition to cardiovascular and renal complications [22].

Unlike other malignancies, MM is characterized by osteolytic bone lesions, not bone growth. These invasive bone lesions result in severe bone aches, osteoporosis and pathologic fractures. Osteolytic bone lesions are critically helpful in the diagnosis of MM, as they are identified in about 80% of MM patients through magnetic resonance imaging and computerized tomography [76]. Further, bone marrow incursion by malignant plasma cells leads to anemia, immunosuppression and recurrent infections [77]. Bone marrow biopsy to detect clonal bone marrow plasma cells is essential for the diagnosis and staging of MM [17]. Other standard confirmatory tests include blood calcium and serum creatinine levels, which are often high in MM patients [17]. International Myeloma Working Group (IMWG) published diagnostic benchmarks for MM and its differential diagnosis, including Non-IgM MGUS, IgM MGUS, Light chain MGUS, SMM and solitary plasmacytoma (Figure 1) [11]. The diagnostic criteria depend primarily on the levels of serum or urinary monoclonal protein and the percentage of clonal bone marrow plasma cells. In addition, evidence of end-organ failure (CRAB criteria): hypercalcemia, renal insufficiency, anemia or bone lesions are fundamental for MM diagnosis. In 2014, the IMWG added three critical features to CRAB, which assisted in the early diagnosis and management of MM cases [11].

Management of MM patients depends on several factors, such as the stage, risk stratification, age and eligibility for transplantation therapy. In the case of low-risk patients, initial therapy includes four cycles of Velcade, Revlimid and Dexamethasone (VRD) followed by autologous stem-cell transplant (ASCT). However, in transplant-ineligible cases, patients continue 8 to 12 VRD cycles. Following the initial therapy, Lenalidomide and Bortezomib are commonly used as maintenance therapy. The initial therapy for high-risk patients comprises Kyprolis, Revlimid and Dexamethasone (KRD) followed by ASCT. Carfilzomib or Bortezomib are used for maintenance treatment [77].

## 5. miRNA and Malignancy

microRNA (miRNA) is a non-coding RNA molecule, 18 to 25 nucleotides in length [78,79]. miRNA constitutes roughly 1% of the human genome [80]. They are synthesized through several steps that include pre-miRNA transcription as well as post-transcriptional modifications through regulatory enzymes such as Drosha, Dicer, RNA-Induced Silencing Complex [79,81]. Although miRNAs do not code for protein, they were found to regulate protein synthesis through controlling messenger (m)RNA transcription and translation, thus influencing various biological processes such as metabolism, cell proliferation and apoptosis [79,82]. Furthermore, miRNA is essential for negative feedback regulation of about 50% of the protein-coding genes. They control gene expression by direct binding to the 3′ untranslated region of mRNA, leading to its degradation or inhibition of its translation [79,81].

miRNAs were shown to regulate the expression of genes controlling cancer development, e.g., oncogenes and tumor suppressor genes. Therefore, they possess a crucial role in the induction, evolution, propagation and metastasis of various malignancies [83]. miRNA may act as oncomiR or tumor suppressor miRNA if they target tumor suppressor genes or oncogenes, respectively [84]. OncomiRs are over-expressed in tumor cells while tumor suppressor miRNAs are usually down-regulated [85]. Thus, miRNA attracted global attention as possible therapeutic targets in cancer management [86].

Interestingly, the role of miRNA in malignancy exceeds the targeting of oncogenes or tumor suppressor genes to potentially form a cell–cell communication approach [87,88] that affects various biological processes [89] at the receiving sites. This is achieved via the circulating miRNAs which are released from the production site in many forms either free or protein-bounded or stalked in membrane-bound vesicles such as extracellular vehicles (EVs) and exosomes. The process of transportation of circulating miRNAs is influenced by many extracellular factors, including plasma RNase, pH, temperature and other digesting enzymes [90]. Circulating miRNAs could be either vesicle-associated [91,92] or non-vesicle-associated, which are released in into the circulation in a protein-bounded form to protect them against plasma RNase [88,93,94,95].

## 6. Role of miRNA in MM

Several studies investigated the dysregulated miRNAs in MM plasma cells to identify the possible miRNAs involved in the MM pathogenesis and progression. Researchers used different approaches such as microarray, quantitative PCR and next-generation sequencing. The experimental samples included bone marrow clonal plasma cells, serum and urine were isolated from patients at various stages of the disease in addition to MM cell lines [96,97]. Interestingly, several miRNAs were dysregulated in MM and its asymptomatic pre-malignant stages. For instance, microarray analysis of samples obtained from both MGUS and MM patients revealed that miR-181a/b, miR-21 and miR-106b/25 were over-expressed in MGUS cases. Further, miR-181a/b, miR-32, miR-17-92, miR-21 and miR-106b/25 were up-regulated in both MM and MGUS patients. However, miR-32 and miR-17/92 were only detected only in MM patients, indicating the potential role of these miRNA in the evolution and transformation of MGUS to MM [98]. Being transported through exosomes, circulating miRNAs were found to contribute to the biogenesis and progression of MM, including tumor survival, proliferation [99,100], malignant transformation of the nearby normal cells [101]. Further, circulating miRNAs were found to enhance drug resistance, osteolysis and angiogenesis [102,103].

In addition to targeting oncogenes or tumor suppressor genes, miRNAs have been shown to have a significant impact on the various biological processes in MM cells, including proliferation, differentiation and angiogenesis in addition to the modulation of the bone marrow microenvironment. For instance, miRNAs can potentially regulate DNA methylation as previously discussed along with other crucial cellular pathways that control cellular proliferation, migration and apoptosis [104,105,106,107,108]. Additionally, miRNAs mediate substantial changes in the bone marrow microenvironment in MM patients. These changes include modulating bone marrow stromal cells to adhere to MM cells resulting in cytokine and growth factor secretion as well as activation of multiple genes and signaling pathways thus, enhancing tumor growth and drug resistance [109,110,111,112]. MM pathogenesis could generally be explained through interactions between MM cells and bone marrow stroma through circulating miRNAs, which was shown to preserve tumor load and enhance metastasis [113,114]. Furthermore, several miRNAs were found to enhance tumor growth by stimulating and augmenting the angiogenesis process through various mechanisms [115,116,117].

### 6.1. OncomiRs and Their Therapeutic Potentials in MM

Several miRNAs are up-regulated in MM to enhance tumor proliferation and growth (Table 1). Therefore, targeting these miRNAs showed remarkable anti-tumor activities. Thorough experiments and investigations revealed various mechanisms through which miRNAs could enhance MM growth and propagation. In this review, we discussed the different pathways targeted by miRNAs resulting in phenotypic and functional changes in addition to their potentials as therapeutic approaches.

#### 6.1.1. Modulation of the Bone Marrow Microenvironment

MM growth requires a specific interaction between bone marrow stromal cells (BMSCs) and MM cells in the bone marrow to generate a favorable microenvironment for MM cell proliferation and survival. To sustain this environment, BMSCs release EVs that contain particular proteins and miRNAs. For example, miR-10a was over-expressed in EVs while it was down-regulated in BMSCs, suggesting that miR-10a was packaged into EVs and released into the bone marrow. Interestingly, inhibition of EV release led to the suppression of cell proliferation and the initiation of apoptosis in BMSCs. Moreover, the transfection of MM cells with miR-10a isolated from BMSCs improved their cell proliferation [118]. miR-181a was remarkably overexpressed in MM cells, where it regulates neuro-oncological ventral antigen-1 (NOVA1) expression. The silencing of miR-181a resulted in a substantial promotion of cellular apoptosis. Furthermore, miR-181a inhibitor reduced the expression of NOVA1 and inhibited tumor growth in vivo. Consistently, miR-181 mimics induced opposite effects [119].

Bone marrow clonal plasma cells were found to up-regulate both miR-27b-3p and miR-214-3p in fibroblasts through the release of exosomes containing WW and C2 Domain Containing 2 (WWC2) protein that enhances the transition of MGUS to MM. This further emphasizes the role of malignant plasma cells in modifying the bone marrow microenvironment by reprogramming fibroblasts’ behavior [120]. Interleukin (IL)-17 producing CD4+ T cells (Th17) are essential for MM growth and bone injury associated with osteoclast activity. Excitingly, inhibition of miR-21 in naive T cells suppressed Th17 differentiation in vitro, thus impairing tumor growth. Mechanistically, blocking of miR-21 resulted in the up-regulation of Signal transducer and activator of transcription (STAT)-1/-5a-5b and impairment of STAT3 pathways [121].

#### 6.1.2. Enhancing Cellular Proliferation and Tumor Growth

Over-expression of miR-27b-3p and miR-214-3p induced proliferation and apoptosis resistance in MM fibroblasts through activation of F-box and WD repeat domain containing 7 (FBXW7) and Phosphatase and tensin homolog (PTEN)/protein kinase B (AKT)/Glycogen synthase kinase (GSK)3 pathways. Consistently, transfection of miR-27b-3p and miR-214-3p inhibitors resulted in suppressing the anti-apoptotic factors, induced myeloid leukemia cell differentiation protein MCL1 [120]. miR-221/222 were found to down-regulate other tumor suppressor genes in MM [122]. miR-221/222 were highly expressed in specific subgroups of MM, and its down-regulation led to inhibition of MM growth in both in vitro and in vivo [123]. In addition, miR-221/222 activates the following pathways: p27Kip1, p57Kip2, PTEN and p53 up-regulated modulator of apoptosis (PUMA) [123]. miR-21 was found to be vital for MM growth and development through activation of pro-survival signaling and targeting tumor suppressor genes such as Ras homolog (Rho)-B and PTEN. Furthermore, anti-miR-21 triggered an over-expression of the previously mentioned tumor suppressor genes [124].

#### 6.1.3. Promoting Cell Migration and Metastasis

Over-expression of miR125a-5p was observed in MM cells [125]. Remarkably, anti-miR-125a-5p significantly reduced cell migration and proliferation in addition to enhancing cellular apoptosis through p53 pathway activation [126]. In MM cells, miR-21 transcription was not regulated by IL-6. As a result, the dysregulated miR-21 enhanced the malignant transformation of plasma cells [97]. miR-19b and miR-20a are crucial oncomiRs that are up-regulated in MM plasma cells. miR-19b/20a enhanced cell proliferation and migration as well as inhibited cell apoptosis in MM. Transfection of miR-19b/20a resulted in the down-regulation of PTEN protein (a tumor suppressor protein with anti-proliferation and pro-apoptosis effects). Lentivirus-mediated delivery of miR-20a boosted tumor growth; therefore, it may resemble a potential therapeutic target [127]. miR-27 was overexpressed in MM cases when compared with control, which was associated with shorter overall survival. miR-27 mimics boosted cell proliferation, migration and invasion through targeting Sprouty homolog 2 (SPRY2). Meanwhile, anti-miR-27 resulted in contrasting effects. miR-27 inhibition showed anti-tumor effects on MM cells [128].

#### 6.1.4. Boosting Cell Viability and Inhibiting Apoptosis

Researchers reported that miR-106b, miR-25 and miR-93 were over-expressed in MGUS and MM cases [98]. Intriguingly, anti-miR-106b/25 reduced the cell viability and induced apoptosis in MM plasma cells by inhibiting p38/mitogen-activated protein kinase (MAPK) dependent signaling pathway [129]. Similarly, miR-19b/20a inhibited cell apoptosis MM cells through targeting PTEN pathway [127]. Furthermore, miR-214-3p increased resistance against apoptosis through targeting the apoptotic FBXW7 and PTEN/AKT/GSK3 pathways [120].

#### 6.1.5. Fostering Drug Resistance

miR-221/222 was over-expressed in plasma cells of MM patients and was found to mediate inhibition of autophagy, which in turn promotes dexamethasone resistance. miR-221/222 targeted autophagy-related gene 12 (ATG12) and p27kip (p27)-mammalian target of rapamycin (mTOR) pathway to reduce autophagy. In fact, Dexamethasone treatment reduced the expression of miR-221/222, thus stimulating ATG12/p27-mTOR pathways and inducing cell apoptosis [130].

### 6.2. Tumor Suppressor miRNAs and Their Therapeutic Potentials in MM

Multiple studies found several miRNAs to be down-regulated in MM plasma cells. These miRNAs possess a pivotal role in the suppression of tumor growth through various pathways (Table 2). Tumor suppressor miRNAs act via inhibiting essential oncogenes, among other mechanisms, to reduce tumor growth.

#### 6.2.1. Inhibiting Cellular Proliferation, Cell Cycle and Tumor Growth

miR-26a was down-regulated in MM plasma cells. Alternatively, induced overexpression of miR-26a reduced proliferation and migration as well as prompted apoptosis in MM cell lines. CD38 protein was found to be targeted and down-regulated by miR-26a. Therefore, CD38 protein was indicated to be responsible for the activation of cell proliferation and the restriction of cell apoptosis. As a result, Daratumumab, the first anti-CD38 drug, was developed as a monotherapy and in combination with other drugs to treat MM cases [131]. miR-29b expression was significantly reduced in MM cells. Thereby, ectopic up-regulation of miR-29b hindered cell proliferation and prompted cycle arrest in malignant plasma cells. miR-29b was shown to target the Forkhead box protein P1 (FOXP1) pathway to induce its effects. The restoration of FOXP1 diminished miR-29b-induced pro-apoptosis and anti-proliferation activities [132]. The expression of miR-489 was significantly reduced in MM cell lines. miR-489 acts as a tumor suppressor gene through reducing cell proliferation and viability of MM cells. Additionally, miR-489 inhibited glucose uptake, thus ATP production. Lactate dehydrogenase-A (LDHA) was identified as a potential target of miR-489. Taken together, a reduction in ATP production and cell proliferation through targeting LDHA was responsible for the inhibitory effects of miR-489 on MM cells [133].

#### 6.2.2. Enhancing Apoptosis and Decreasing Cell Viability

miR-155 expression was significantly reduced in MM cases, thus indicating a tumor suppressor role. miR-155 replacement induced pro-apoptotic and anti-proliferative effects in the MM cell line [134]. Mitochondrial RNA processing endoribonuclease (RMRP) was up-regulated, whereas miR-34a-5p was down-regulated in MM cell lines. Over-expression of RMRP expression enhanced cell proliferation of MM cell lines. Consistently, RMRP knockdown induced apoptosis in the same cells. The silencing of miR-34a-5p was associated with high RMRP levels. Moreover, high expression of miR-34a-5p inhibited proliferation and fostered apoptosis, which indicates that RMRP acts as a miR-34a-5p MASK to boost cell proliferation and suppress cell apoptosis [135]. miR-15a and miR-16-1 were down-regulated in MM cases. Furthermore, malignant plasma cells transfected with miR-15a/16-1 showed cellular apoptosis and repressed tumor growth. Both miR-15a and miR-16-1 were found to inhibit tumor survival pathways, including mitogen-activated protein (MAP)-kinases, AKT serine/threonine-protein-kinase, NF-κB-activator MAP3KIP3 and ribosomal-protein-S6 [136].

Similarly, miR-125b was down-regulated in MM patients [137] by cancer-secreted growth factors such as tumor necrosis factor (TNF) and insulin growth factor (IGF)-1 [138,139]. Restoration of miR-125b through the replacement approach showed a remarkable inhibitory effect on malignant plasma cells, which was achieved by repressing interferon regulatory factor (IRF)4 addiction, which is essential for MM growth and development [138,139]. Interestingly, miR-125b was found to up-regulate miR-34a, which in turn inhibits the IL-6 receptor/STAT3/miR-34a feedback loop. As a result, these pathways activate cell death machinery in MM [140]. The up-regulation of miR-33b resulted in significant apoptosis of malignant plasma cells, thus suppressing cancer growth and enhancing survival rates [141]. Interestingly, miR-33b targeted a serin/threonine kinase known as PIM-1, which blocks the linkage between BCL2 associated agonist of cell death (Bad) and Bcl2/l-xl to inhibit apoptosis [141]. p53, a potent tumor suppressor gene, was found to have a critical role in protecting against MM development. Several miRNAs were reported to be engaged in regulating p53 during the disease, such as miR-194-2/192 and miR-215/194-1 families [142].

miR-34a mimics inhibited tumor growth in MM mice models through activation of apoptotic pathways and suppression of pro-survival signaling, including cell division protein kinase (CDK)6, BCL2 and NOTCH1 [143]. Additionally, miR-34a was found to reduced plasma cell proliferation by repressing transforming growth interaction factor 2 (TGIF2) [144]. Similarly, miR-29b mimics down-regulated proliferative and anti-apoptotic pathways in MM cells such as MCL-1, CDK6 and SP1 [145] in addition to other epigenetic regulators [146] including DNA methyltransferase three alpha/beta (DNMT3A/B) [147] and Histone Deacetylase 4 (HDAC4) [148]. Additionally, mir-29b increased the MM sensitivity towards Bortezomib [145,149].

#### 6.2.3. Increasing Sensitivity to Drugs

miR-155 was down-regulated in Bortezomib-resistant MM patients suggesting a possible role of miR-155 in Bortezomib resistance. miR-155 replacement augmented Bortezomib therapeutic efficacy. Researchers suggested that the anti-tumor properties of miR-155 were attributed to inhibiting proteasome subunit gene, PSMβ5 [134]. miR-192/215/194 were found to be down-regulated in MM cases [150]. The expression of these miRNAs was usually accompanied by an activation of the p53 pathway, which is associated with low Mouse double minute (MDM)2 levels. Consistently, the replacement of miR-192, miR-215 and miR-194 augmented the therapeutic efficacy of MDM2 inhibitors through enhancing p53 pathway [150]. miR-214 inhibits p28/gankyrin, which are oncogenes that suppress p53 by binding to the MDM2/HDM2 complex. Therefore, miR214 replacement improves p53 signaling by blocking gankyrin oncoproteins [151]. The expression of miR-520g/520h was significantly low in Bortezomib-resistant MM cell lines. Moreover, the up-regulation of miR-520g and miR-520h impeded cell viability and the expression of the homologous recombination-related protein (Rad51) in Bortezomib-resistant MM cells by targeting Apurinic/apyrimidinic endonuclease (APE)1 in vitro, in addition to repressing tumor growth in vivo [152].

#### 6.2.4. Hindering Survival and Genomic Instability

Activation of DNA ligase III (LIG3) is critical for the survival and genomic inconsistency of MM cells. The up-regulation of LIG3 mRNA is associated with more advanced MM and shorter survival. Interestingly, miR-22-3p was identified as an efficient inhibitor of LIG3, which, in turn, enhanced DNA damage in MM cells [153]. miR-125a was down-regulated in MM cell lines. It functions as a tumor suppressor gene by reducing cell viability and colony-forming activity. miR-125a targets ubiquitin-specific peptidase 5 (USP5) mRNA, an oncoprotein that enhances cellular deubiquitination and proteolysis. Highly expressed miR-125a significantly repressed tumor growth of MM and lowered USP5 expression in vivo [154]. miR-101-3p was down-regulated and survivin (BIRC5) was up-regulated in MM cells. Notably, miR-101-3p was found to target and down-regulate survivin, which reduces the cell viability of malignant plasma cells. Moreover, anti-miR-101-3p was associated with a high expression of survivin [155]. miR-155 was down-regulated in drug-resistant MM through regulating CD47. Up-regulation of miR-155 inhibited CD47 expression on the plasma cell surface, thereby promoting phagocytosis of MM cells by macrophages. Moreover, miR-155 enhanced the sensitivity of drug-resistant MM cells to Bortezomib through targeting Tumor Necrosis Factor Alpha Induced Protein 8 (TNFAIP8), an oncoprotein which negatively inhibits apoptosis [156].

#### 6.2.5. Blocking Angiogenesis

Angiogenesis is exceptionally critical for tumor growth in MM [157]. Several miRNAs were involved in the regulation of endothelial cells proliferation and new blood vessel formation. For instance, the replacement of miR-15a/16 inhibited vascular endothelial growth factor (VEGF) expression, thus suppressing the capillary formation and tumor growth [158]. miR-199a-5p down-regulated the gene expression of angiogenic factors such as VEGF-A, hypoxia-inducible factors (HIF)-1α, IL-8 and fibroblast growth factor (FGF)-b [116]. Moreover, miR-199a-5p was shown to regulate invasion and metastasis of MM [159]. Administration of miR-199a-5p mimic was proven to inhibit malignant cells’ chemotaxis pathways, including metalloproteinase (MMP)2, Vascular cell adhesion protein (VCAM)-1, Discoidin domain receptor (DDR)1 and Intercellular Adhesion Molecule (ICAM)-1 [116]. Other pathways for cellular migration incorporate Wnt pathway and its regulatory factors, e.g., B-cell CLL/lymphoma (BCL)9 and β-catenin [160]. Intriguingly, miR-30-5p impeded proliferation and migration of MM plasma cells through inhibiting the BCL9 pathway [160].

## 7. miRNAs Therapeutic Strategies in MM

Being involved in the development and the evolution of MM, miRNAs could represent potential therapeutic targets. Two strategies are currently being investigated: inhibition of up-regulated oncomiRs and replacement of down-regulated tumor suppressor miRNAs. Some of the challenges in miRNA therapeutic strategies include miRNA stability, selective cellular uptake by target cells through an effective delivery system, possible off-target and unwanted toxicities and the activation of innate immune responses [161,162].

### 7.1. Inhibition of oncomiRs

On the other hand, specific miRNAs are up-regulated in MM to enhance proliferation, growth and migration of malignant cells. Consequently, scientists are currently developing several approaches to target these miRNAs. For instance, antisense miRNA inhibitors (antagomirs) bind exclusively to the sense miRNA resulting in its inhibition [163]. Moreover, locked nucleic acid linked to a phosphorothioate backbone is commonly used to augment the stability and affinity of antagomirs to their target miRNAs [164,165]. Other approaches include miRNA sponges, which are transcripts that have several binding sites that preclude binding sites of the oncomiRs to mRNA [166]. Similarly, MASK, a synthesized oligonucleotide complementary to the binding sites of miRNA, blocks its interaction with mRNA [167].

miR-17-92 cluster is a polycistron encoding 6 miRNAs (miR-17, miR-18a, miR-19a, miR-19b-1, miR-20a and miR-92-1) confer tumorigenicity in MM, regulated by c-Myc, an oncogenic transcription factor [168]. The miR-17-92 cluster is encoded by MIR17HG at 13q31.3 [169]. MIR17PTi is a novel locked nucleic acids gapmeR antisense oligonucleotide to induce degradation of MIR17HG primary transcripts (pri-mir-17-92) and thus resulted in the down-regulation of miR-17-92 miRNAs [170]. Furthermore, MIR17PTi treatment was reported to inhibit malignant plasma cell proliferation and stimulate cellular apoptosis, suggesting the potential of pri-miRNA therapeutics in cancer therapy [170].

### 7.2. Replacement of Tumor Suppressor miRNAs

Some of the miRNAs that are down-regulated in different categories of MM may have potential inhibitory effects on tumor growth and metastasis. Therefore, identification of these miRNAs and their biological functions may help to develop powerful therapeutic tools through suppression of pro-survival conditions and progression of MM. One such example is miR-34a, which is known to act as a tumor suppressor in numerous cancers. In fact, MRX34, a liposomal miR-34a mimic, is the first human phase I clinical trial of miRNA cancer therapy in all patients with advanced solid tumors [171]. Although the trial was closed early due to serious immune-mediated adverse effects that resulted in the death of four patients, the dose-dependent modulation of relevant target genes provides proof-of-concept for miRNA-based cancer therapy [172]. The combination treatment with miR-34a has been shown to enhance the anti-tumor activity of other anti-cancer agents (γ-secretase inhibitor, sirtinol and zoledronic acid) in MM cells, through various mechanisms [173]. The γ-secretase inhibitor enhanced miR-34a-dependent anti-tumor effects by activating an extrinsic apoptotic pathway, whereas the combination of miR-34a and sirtinol, induced the activation of an intrinsic apoptotic pathway [173].

To replace the down-regulated miRNAs, different tools, including viral vectors, non-viral vectors (inorganic compounds and lipid-based carries) and miRNAs mimics can be used [174,175]. However, most MM cells are transfection-resistant. A non-viral nanoparticulate transfection system, poly(glycidol)-based nanogels with covalently bound cell-penetrating peptide TAT (transactivator of transcription) has shown to efficiently delivered and released miR-34a into transfection-resistant OPM-2 MM cells [176]. The delivery of miR-34a resulted in significant down-regulation of known target genes (Notch1, Hey1, Hes6 and Hes1), suggesting the nanogel constructs offer a new tool to enhance gene delivery [176].

## 8. Circulating miRNAs as Biomarkers for MM Diagnosis

Since miRNAs have been found in body fluids such as plasma, serum, saliva, urine and semen, circulating miRNAs have been proposed as novel disease biomarkers that may aid in diagnosis, prognosis and monitoring of treatment response [177,178,179,180]. The circulating miRNAs were stable when refrigerated or frozen for up to 72 h and at room temperature for 24 h, making them suitable biomarkers [181,182].

Serum analysis revealed a combination of up-regulated miR-34a and down-regulated let-7e could distinguish MM from control with a sensitivity and specificity of 80%, and MGUS with a sensitivity and specificity of over 90% (Table 3) [183]. Other suggested markers include increased plasma miR-125b-5p, serum miR-29a, serum miR-4449, and decreased serum miR-30d and miR-203 levels [184,185,186,187,188]. In addition, the miR-125b-5p level was associated with extramedullary infiltration and was significantly higher in stage III patients compared to stage I/II patients [184]. Similarly, plasma miR-483-5p level in MM patients was also found to be correlated with ISS stage [189].

Apart from being a diagnostic marker, numerous mRNAs have been suggested for diagnosis of survival prediction in MM patients. Low levels of serum miR-744 and let-7e were associated with shorter overall survival and remission of MM patients [183], while high levels of plasma miR-483-5p, serum miRNA-720 and miRNA-1246 were associated with shorter progression-free survival, indicating poor prognosis [189,190].

## 9. Conclusions

This review summarized various miRNAs with the role of tumor suppressor and oncomiR in MM. In addition, miRNAs are also involved in treatment resistance in MM patients. Currently, the full complement of miRNAs involved in the pathogenesis of MM has yet to be fully elucidated. Further studies on miRNA dysregulation in MM may provide novel sensitive diagnostic markers, therapeutic options for MM, as well as to resolve treatment resistance in patients.

## Figures and Tables

**Figure 1 ijms-21-07539-f001:**
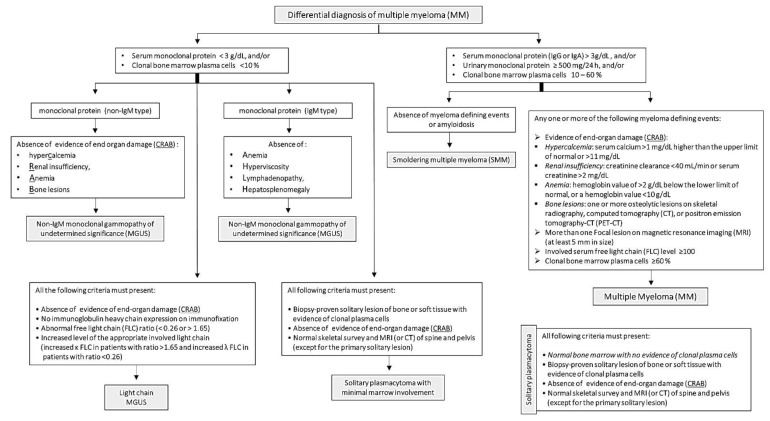
Diagnostic criteria of multiple myeloma and its differential diagnosis published by the International Myeloma Working Group.

**Table 1 ijms-21-07539-t001:** miRNAs acting as oncomiRs in multiple myeloma (MM) and their potential targets.

miRNA	Mechanisms of Action	Targets	Reference
miR-10a	↑ cell proliferation	-	[118]
	↓ apoptosis		
miR-19b	↑ cell proliferation and migration	PTEN protein	[127]
	↓ apoptosis		
miR-20a	↑ cell proliferation and migration	PTEN protein	[127]
	↓ apoptosis		
miR-21	↓ CD4+ T cells (Th17) differentiation	STAT-1/-5a-5b and STAT3	[121]
	↑ pro-survival signaling	Rho-B and PTEN	[124]
miR-25	↑ cell viability	p38/MAPK	[129]
	↓ apoptosis		
miR-27	↑ cell proliferation, migration and invasion	SPRY2	[128]
miR-27b-3p	↑ proliferation and apoptosis resistance	FBXW7 and PTEN/AKT/GSK3	[120]
miR-93	↑ cell viability	p38/MAPK	[129]
	↓ apoptosis		
miR-106b	↑ cell viability	p38/MAPK	[129]
	↓ apoptosis		
miR-181a	↓ apoptosis	NOVA1	[119]
miR-125a-5p	↑ cell proliferation and migration	-	[125,126]
	↓ apoptosis		
miR-214-3p	↑ proliferation and apoptosis resistance	FBXW7 and PTEN/AKT/GSK3	[120]
miR-221	↓ autophagy	ATG12 and p27/mTOR	[130]
	↑ Dexamethasone resistance		
	↑ tumor growth	p27/Kip1, p57Kip2, PTEN and PUMA	[123]
miR-222	↓ autophagy	ATG12 and p27/mTOR	[130]
	↑ Dexamethasone resistance		
	↑ tumor growth	p27/Kip1, p57/Kip2, PTEN and PUMA	[123]

↑ increase; ↓ decrease.

**Table 2 ijms-21-07539-t002:** miRNAs acting as tumor suppressor genes in MM and their potential targets.

miRNA	Mechanisms of Action	Targets	Reference
miR-15a, miR-16	↓ capillary formation, tumor growth	VEGF	[158]
	↑ apoptosis	MAP-kinases, AKT serine/threonine-protein-kinase, NF-κB-activator MAP3KIP3 and ribosomal-protein-S6	[136]
miR-22-3p	↓ survival, genomic instability	DNA ligase III	[153]
miR-26a	↓ cell proliferation, migration	CD38	[131]
	↑ apoptosis		
miR-29b	↑ apoptosis	MCL-1, CDK6 and SP1	[145]
	↓ cell proliferation	DNMT3A/B and HDAC4	[147,148]
	↑ sensitivity to Bortezomib	MCL-1, CDK6 and SP1	[145,149]
	↓ cell proliferation	FOXP1	[132]
	↑ cell cycle arrest		
miR-30-5p	↓ cell proliferation, migration	BCL9	[160]
miR-33b	↓ linkage between Bad and Bcl2/l-xl	PIM-1	[141]
	↑ apoptosis		
miR-34a	↓ tumor growth	IL-6 receptor/ STAT3	[140]
	↑ apoptosis	CDK6, BCL2 and NOTCH1	[143]
	↓ pro-survival signaling		
miR-34a-5p	↓ proliferation	RMRP	[135]
	↑ apoptosis		
miR-101-3p	↓ cell viability	survivin (BIRC5)	[155]
miR-125a	↓ cell viability, colony-forming activity	USP5	[154]
miR-125b	↓ tumor growth	IRF4	[138,139]
miR-155	↑ pro-apoptotic, anti-proliferative effects	proteasome subunit gene, PSMβ5	[134]
	↑ Bortezomib therapeutic sensitivity		
	↑ phagocytosis of MM cells by macrophages	CD47	[156]
	↑ sensitivity of drug-resistant MM cells to Bortezomib	TNFAIP8	[156]
miR-192	↓ cell proliferation	TGIF2	[144]
miR-192, miR-194, miR-215	Augmented the therapeutic efficacy of MDM2 inhibitors	p53 and MDM2	[150]
miR-199a-5p	↓ capillary formation, tumor growth	VEGF-A, HIF-1α, IL-8 and FGFb	[116]
	↓ plasma cells chemotaxis	MMP2, VCAM-1, DDR1 and ICAM-1	[116]
miR-214	↓ tumor growth	p53 and p28/gankyrin	[151]
miR-489	↓ cell proliferation, viability	LDHA	[133]
	↓ glucose uptake, ATP production		
miR-520g,h	↓ cell viability	Rad51 and APE1	[152]

↑ increase; ↓ decrease.

**Table 3 ijms-21-07539-t003:** miRNAs acting as diagnostic markers in MM.

Sample	miRNAs	Expression Changes in MM Patients	Sensitivity	Specificity	Reference
Plasma	miR-34a	↑	81%	87%	[183]
	let-7e	↓			
	miR-125b-5p	↑	86%	96%	[184]
	miR-490-3p	↑	60%	85%	
Serum	miR-29a	↑	88%	70%	[185]
	miR-203	↓	83%	83%	[186]
	miR-4449	↑	79%	91%	[187]
	miR-30d	↓	89%	63%	[188]
	miR-483-5p	↑	58%	90%	[189]
	miR-20a	↓	63%	85%	

↑ increase; ↓ decrease.

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
