# Peer review of "Role of microRNAs in Diagnosis, Prognosis and Management of Multiple Myeloma"

_ijms, 2020, doi:10.3390/ijms21207539_

Round 1

Reviewer 1 Report

The article by Soliman et al :”Role of microRNAs in Multiple Myeloma” to be considered for publication in Int J of Mol Sciences has the overall ambition to give an overview of the present literature on this subject. Since Pichiorri et al. published the first comprehensive microRNA expression profile covering normal PCs, MGUS, MM samples and MM cell lines revealing an upregulated microRNA signature in MM, the field has extended with numerous articles describing oncogenic properties of miRNAs. In parallel several studies have reported underexpressed miRNA profiles, thus with putative tumor suppressor functions in MM.  Although the authors list the findings in several of these publications, this list and attached tables contribute very little to our understanding of the field.

To be considered for publication as a review - a major revision should be made:

A) In general the text is far too descriptive to be interesting as a review.

B) The introduction on the disease multiple myeloma may take into account other levels of epigenetic regulation to set the proper frame to miRNAs.

C) The section 5 is at present subsectioned in OncomiRs and Tumorsuppressor miRNAs. Rather than listing the literature findings in an order lacking rational for the reader, the authors should :

1) discuss results presented in the literature only with correlative vs which findings show causative relationship between miRNA expression, function and phenotypic consequences.

2) divide the sections of oncomiRs and TS miRNAs further according to such a discussion.

3) the action of miRNAs may not be restricted to the repression of selected single targets since but rather show a major impact on overall biological processes i.e. DNA methylation, angiogenesis, bone modulation etc.  This should be discussed in separate sections.

4) the discussion and potential problems with transportation of miRNA between the site of production and site of function is not addressed. A review should include a discussion on this, not only refer to findings on EVs containing miRNAs.

Overall the impression is at present, unfortunately, that this article is the introduction to a PhD thesis rather than a comprehensive review. By the suggested changes on the structure and a thourough discussion on the actual potential of these results the review might be considered for publication.

Author Response

Reviewer 1

The article by Soliman et al :”Role of microRNAs in Multiple Myeloma” to be considered for publication in Int J of Mol Sciences has the overall ambition to give an overview of the present literature on this subject. Since Pichiorri et al. published the first comprehensive microRNA expression profile covering normal PCs, MGUS, MM samples and MM cell lines revealing an upregulated microRNA signature in MM, the field has extended with numerous articles describing oncogenic properties of miRNAs. In parallel several studies have reported underexpressed miRNA profiles, thus with putative tumor suppressor functions in MM.  Although the authors list the findings in several of these publications, this list and attached tables contribute very little to our understanding of the field.

To be considered for publication as a review - a major revision should be made:

  1. A) In general the text is far too descriptive to be interesting as a review.
  2. B) The introduction on the disease multiple myeloma may take into account other levels of epigenetic regulation to set the proper frame to miRNAs.

We appreciate the comments by the reviewer and rectified the manuscript. In the revised version, a new section (section 3) has been added to cover the epigenetics of MM

  1. C) The section 5 is at present subsectioned in OncomiRs and Tumorsuppressor miRNAs. Rather than listing the literature findings in an order lacking rational for the reader, the authors should:

1) discuss results presented in the literature only with correlative vs which findings show causative relationship between miRNA expression, function and phenotypic consequences.

2) divide the sections of oncomiRs and TS miRNAs further according to such a discussion.

For 1) and 2), subsections were added in section 6 as [per suggestion to cover points proposed in number (1).

3) the action of miRNAs may not be restricted to the repression of selected single targets since but rather show a major impact on overall biological processes i.e. DNA methylation, angiogenesis, bone modulation etc.  This should be discussed in separate sections.

Parts were added to section 5 and 6 to cover the suggested points.

4) the discussion and potential problems with transportation of miRNA between the site of production and site of function is not addressed. A review should include a discussion on this, not only refer to findings on EVs containing miRNAs.

A paragraph was added in section 5 to cover the suggested points.

Overall the impression is at present, unfortunately, that this article is the introduction to a PhD thesis rather than a comprehensive review. By the suggested changes on the structure and a thorough discussion on the actual potential of these results the review might be considered for publication.

We made it a thorough discussion in the revised version. More than 70 references were added to the text in order to cover the critical points suggested by the honorable reviewer.

Reviewer 2 Report

Soliman and Collegues have properly revised the available Literature related to the role of microRNAs (miRNAs) within the specific context of multiple myeloma (MM). The most recently dereuglated miRNAs have been discussed as for their relevance in supporting the pathogenesis of MM. Of note, Authors have also discussed on the role of miRNAs as potential therapeutic targets for MM patients.

The manuscript is well written.

Minor comment: Authors have well addressed the important role of the bone marrow (BM) milieu in MM; and have also referred to the crucial role of exosome-tranfer from the BM to the tumor clone. The present reviewer suggests including the first reserach paper describing exosome-mediated transfer of miRNAs from the BM microenvironment to the clonal plasma cells (J CLIN INVEST, 2013;123:1542-55).

Author Response

Reviewer 2

Soliman and Collegues have properly revised the available Literature related to the role of microRNAs (miRNAs) within the specific context of multiple myeloma (MM). The most recently dereuglated miRNAs have been discussed as for their relevance in supporting the pathogenesis of MM. Of note, Authors have also discussed on the role of miRNAs as potential therapeutic targets for MM patients.

The manuscript is well written.

Minor comment: Authors have well addressed the important role of the bone marrow (BM) milieu in MM; and have also referred to the crucial role of exosome-tranfer from the BM to the tumor clone. The present reviewer suggests including the first reserach paper describing exosome-mediated transfer of miRNAs from the BM microenvironment to the clonal plasma cells (J CLIN INVEST, 2013;123:1542-55).

We thank the reviewer for the comments. The suggested reference has been added (Refer to the new reference No.112).

Round 2

Reviewer 1 Report

The authors have improved the text as suggested. By the suggested changes on the structure the review might  be considered for publication.

To consider for the authors is that to contribute greatly to our current understanding of the field, a review should preferable include not only listing of results but also a discussion of these results from a new standpoint taken by the authors.